# A helminth-derived suppressor of ST2 blocks allergic responses

Francesco Vacca[1], Caroline Chauché[1], Abhishek Jamwal[2], Elizabeth C Hinchy[3], Graham Heieis[4], Holly Webster[4], Adefunke Ogunkanbi[5], Zala Sekne[2], William F Gregory[1,6], Martin Wear[7], Georgia Perona-Wright[4], Matthew K Higgins[2], Josquin A Nys[3], E Suzanne Cohen[3], Henry J McSorley[1,5]*

[1]Centre for Inflammation Research, University of Edinburgh, Queen's Medical Research Institute, Edinburgh, United Kingdom; [2]Department of Biochemistry, University of Oxford, Oxford, United Kingdom; [3]Bioscience Asthma, Research and Early Development, Respiratory & Immunology, BioPharmaceuticals R&D, AstraZeneca, Cambridge, United Kingdom; [4]Institute of Infection, Immunity and Inflammation, University of Glasgow, Glasgow, United Kingdom; [5]Division of Cell Signalling and Immunology, School of Life Sciences, Wellcome Trust Building, University of Dundee, Dundee, United Kingdom; [6]Division of Microbiology & Parasitology, Department of Pathology, University of Cambridge, Tennis Court Road, Cambridge, United Kingdom; [7]The Edinburgh Protein Production Facility (EPPF), Wellcome Trust Centre for Cell Biology (WTCCB), University of Edinburgh, Edinburgh, United Kingdom

**Abstract** The IL-33-ST2 pathway is an important initiator of type 2 immune responses. We previously characterised the HpARI protein secreted by the model intestinal nematode *Heligmosomoides polygyrus*, which binds and blocks IL-33. Here, we identify *H. polygyrus* Binds Alarmin Receptor and Inhibits (HpBARI) and HpBARI_Hom2, both of which consist of complement control protein (CCP) domains, similarly to the immunomodulatory HpARI and Hp-TGM proteins. HpBARI binds murine ST2, inhibiting cell surface detection of ST2, preventing IL-33-ST2 interactions, and inhibiting IL-33 responses in vitro and in an in vivo mouse model of asthma. In *H. polygyrus* infection, ST2 detection is abrogated in the peritoneal cavity and lung, consistent with systemic effects of HpBARI. HpBARI_Hom2 also binds human ST2 with high affinity, and effectively blocks human PBMC responses to IL-33. Thus, we show that *H. polygyrus* blocks the IL-33 pathway via both HpARI which blocks the cytokine, and also HpBARI which blocks the receptor.

*For correspondence:
hmcsorley001@dundee.ac.uk

## Introduction

Intestinal nematodes such as human hookworms are highly successful parasites, infecting almost 500 million people worldwide (*Loukas et al., 2016*). They are associated with immunosuppression (*Bethony et al., 2006*), which in turn may reduce the prevalence of immune-mediated diseases such as asthma. Epidemiological studies indicate that hookworm-infected populations tend to have a lower prevalence of diseases such as allergy and asthma (*Leonardi-Bee et al., 2006*), although analysis of these studies is complicated by readouts of allergic disease, timing, dose and species of parasitic infection, and non-infectious socioeconomic factors (*McSorley et al., 2019*; *Santiago and Nutman, 2016*). Studies in animal models of parasitic infection support this association, with suppression of allergic pathology seen after parasitic infection (*Chenery et al., 2016*; *Obieglo et al., 2018*; *Smits et al., 2007*; *Wilson et al., 2005*). Suppression of allergic responsiveness or tolerization of the host to parasite and bystander allergens has been hypothesised to be through the release of

immunomodulatory secretions from the parasite, which has been supported by numerous studies in mouse models of asthma showing the immunosuppressive capacity of parasite secretions (*Buck et al., 2014*; *Ebner et al., 2014*; *Finlay et al., 2017*; *McSorley et al., 2014*; *McSorley et al., 2012*; *Trujillo-Vargas et al., 2007*). However, until recently, the specific molecules and mechanisms by which parasites achieve this have remained elusive.

The IL-33 pathway has critical importance in type two immune responses both in allergy (*Bønnelykke et al., 2014*) and helminth parasite ejection (*Coakley et al., 2017*; *Hung et al., 2013*), thus it could be a central factor in parasite immunomodulation and the prevention of asthma. IL-33 is an alarmin cytokine, released from epithelial, endothelial and stromal cells on necrotic cell death caused by noxious stimuli, such as allergen or pollutant inhalation, infections and mechanical damage (*Johansson and McSorley, 2019*; *Mahlakõiv et al., 2019*; *Spallanzani et al., 2019*). Recent studies also show that mast cells can also be an important source of IL-33 in *H. polygyus* infection, secreting the cytokine in response to damage signals from epithelial cells (*Shimokawa et al., 2017*). Extracellular IL-33 binds to its receptor complex, consisting of ST2 (*Il1rl1*) and the IL-1 Receptor Accessory Protein (IL1RAcP) (*Günther et al., 2017*). IL-33 receptor signalling induces IL-5 and IL-13 production by type two innate lymphoid cells (ILC2s), IL-6 from mast cells and IFNγ from NK and γδT cells (*Cayrol and Girard, 2018*).

From studies of the murine model intestinal nematode *Heligmosomoides polygyrus*, we recently identified the *H. polygyrus* Alarmin Release Inhibitor (HpARI), a protein secreted by a murine intestinal nematode which binds to DNA and IL-33 in necrotic host cells, tethering the cytokine in the nucleus, preventing its release while simultaneously directly blocking binding to its receptor ST2 (*Osbourn et al., 2017*).

*H. polygyrus* also secretes Hp-TGM, a mimic of host TGF-β which induces regulatory T cells (*Johnston et al., 2017*). Intriguingly, both HpARI and Hp-TGM consist of a string of consecutive atypical Complement Control Protein domains (CCP/Sushi/SCR domains, Interpro IPR000436). As proteins containing CCP domains are greatly expanded in *H. polygyrus* (*Maizels et al., 2018*), and many of these CCP domain-containing proteins are secreted (*Hewitson et al., 2013*) we hypothesised that the CCP domain-containing family represents an immunomodulatory family of proteins.

Here, we identify the *H. polygyrus* Binds Alarmin Receptor and Inhibits (HpBARI), a protein secreted by the parasite which consists of two atypical CCP domains. HpBARI binds and blocks ST2, inhibiting IL-33 responses in a murine model of asthma. During *H. polygyrus* infection, ST2 detection on peritoneal lavage and lung cells was abrogated, which is consistent with the blocking effects of HpBARI. We furthermore identified a close homologue of HpBARI (HpBARI_Hom2) which is able to bind and inhibit the human form of the IL-33 receptor. This study highlights the importance of IL-33 modulation to *H. polygyrus*, expands the family of CCP domain containing protein immunomodulators from the parasite, and identifies a novel agent with potential for treatment of human IL-33-mediated disease.

## Results

### *H. polygyrus* products suppress ST2 detection on immune cells

HpARI and HES were compared for their ability to block *Alternaria* allergen responses in vivo. While both HES and HpARI could suppress eosinophilia and ILC2 responses (*McSorley et al., 2014*; *Osbourn et al., 2017*), we found significant differences in ST2 detection on lung ILC2s. *Alternaria* allergen administration induced increased expression of ST2, and while HpARI reduced levels of ST2 to that of the PBS control (presumably due to blockade of IL-33 signalling) we found that HES suppressed detection of ST2 on ILC2s to levels far below baseline (*Figure 1A–B*). We therefore hypothesised that a HES constituent distinct from HpARI was able to block ST2 directly. An in vitro assay was set up to further test this, using naïve murine lung cells, cultured for 24 hr with HES. Detection of ST2 on lung ILC2s was reduced in a dose-dependent fashion when HES was added (*Figure 1C*).

### The HpBARI protein is the ST2-suppressive factor in HES

HES size and charge fractions were then tested in this 24 hr culture assay, and candidate proteins which correlated with ST2 suppression identified by LC-MS/MS (as used to identify HpARI and Hp-TGM [*Johnston et al., 2017*; *McSorley et al., 2014*; *Figure 2—figure supplement 1A*]).

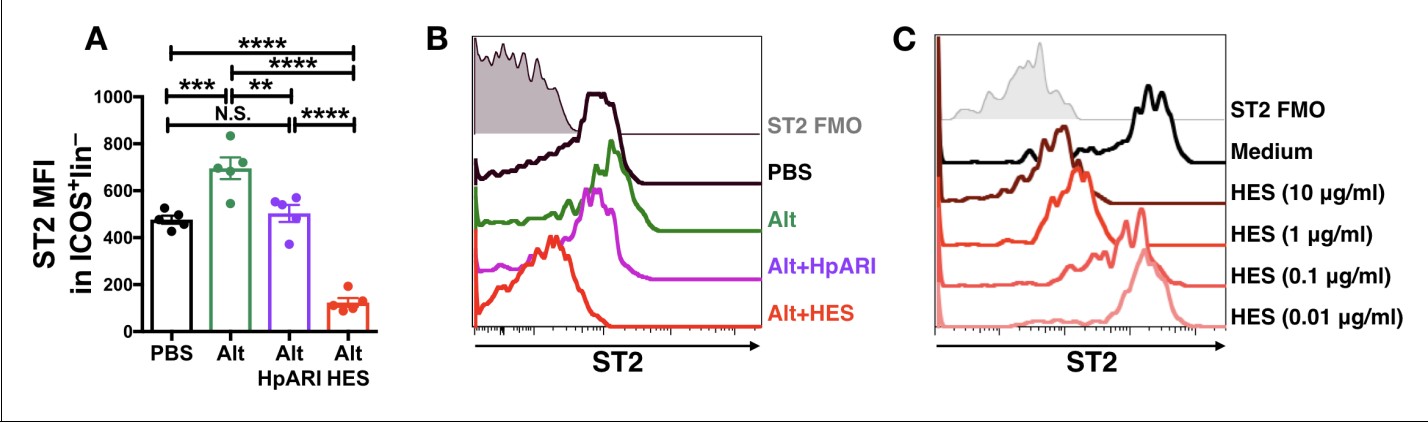

**Figure 1.** HES contains a factor, distinct from HpARI, which suppresses detection of ST2. (A–B) HpARI (5 µg) or HES (10 µg) were coadministered with 25 µg of *Alternaria* allergen by the intranasal route, and lung cell ST2 staining assessed 24 hr later. Geometric mean fluorescence intensity (MFI) of ST2 staining on ILC2 (ICOS⁺lineage⁻CD45⁺) is shown in (A), with representative histograms shown in (B). Representative of 2 replicate experiments, each with 3–5 mice per group. Error bars show SEM. (C) Naïve murine lung cells were cultured for 24 hr in the presence of HES at the concentrations indicated, after which ST2 MFI on ILC2 was assessed. Data representative of >3 repeat experiments, n = 3 per group.

Of the 20 candidate proteins which proteomic analysis indicated most accurately correlated with the ST2-suppressive effect (*Figure 2—figure supplement 1B*), two were closely-related genes encoding CCP domain-containing proteins (Hp_I25642_IG17586_L548 and Hp_I25217_IG17161_L558, 87% identity). As both HpARI and Hp-TGM consist of atypical CCP domains, these CCP domain-containing candidates were of particular interest. Unlike previous studies where multiple candidate proteins were tested (*Johnston et al., 2017*; *Osbourn et al., 2017*), only Hp_I25642_IG17586_L548 was selected for further testing, due to its identification as a CCP domain-containing protein, and as it had previously been gene synthesized, expressed and tested for HpARI-like IL-33 binding activity (for which it was negative). A codon-optimised sequence encoding the Hp_I25642_IG17586_L548 protein was cloned into an expression vector with an N-terminal 6-His and myc tag and expressed in Expi293F mammalian cells. The expressed protein was predicted to have a molecular weight of 23 kDa, and ran on an SDS-PAGE gel at around 30 kDa (*Figure 2—figure supplement 1C*).

Purified recombinant Hp_I25642_IG17586_L548 showed a dose-dependent ability to suppress ST2 detection on lung ILC2s in vitro, acting at ~100 fold lower concentration than total HES (*Figure 2A*). Culture of naïve murine bone marrow cells with a combination of IL-2, IL-7 and IL-33 induced production of IL-5, IL-6 and IL-13, all of which were dose-dependently suppressed by the Hp_I25642_IG17586_L548 protein (*Figure 2B–D*). Thus, we named this protein as '*H. polygyrus* Binds Alarmin Receptor and Inhibits' (HpBARI).

## HpBARI is a CCP domain protein

The HpBARI protein consists of two CCP domains, each containing four conserved cysteines and a tryptophan between the third and fourth cysteine residue (*Kirkitadze and Barlow, 2001*; *Figure 2—figure supplement 2A*).

A Wormbase ParaSite BLAST search for HpBARI in the published *H. polygyrus* genome was carried out (*Howe et al., 2016*; *Howe et al., 2017*), identifying a stretch of genomic DNA showing 97% identity to the HpBARI sequence over five exons (*Figure 2—figure supplement 2B*). As expected for a CCP domain protein, each CCP domain was encoded over two exons, with intron/exon boundaries falling between each CCP domain (*Figure 2—figure supplement 2C*; *Johnston et al., 2017*; *Osbourn et al., 2017*).

The C-terminal region of the HpBARI protein appears to be critical for its binding and function, as when HpBARI was expressed with C-terminal purification tags (HpBARI_C), its activity was ablated in in vitro IL-33 suppression assays (*Figure 2—figure supplement 3*).

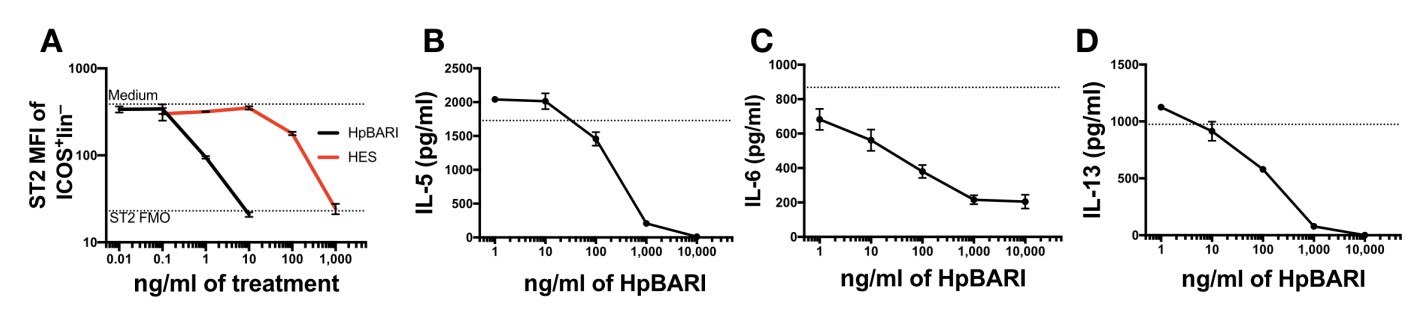

**Figure 2.** HpBARI suppresses ST2 detection, and suppresses IL-33 responses in vitro. (**A**) Naïve murine lung cells were cultured at 37°C overnight with HES or recombinant HpBARI, and ST2 expression measured by flow cytometry. (**B–D**) Naïve murine bone marrow cells cultured for 5 days with IL-2, IL-7 and IL-33 (all at 10 ng/ml) followed by ELISA of cell-free supernatants for IL-5 (**B**), IL-6 (**C**) and IL-13 (**D**). Dotted line indicates levels with IL-2, IL-7 and IL-33 alone. All data are representative of >3 repeat experiments, with three technical replicates per measurement. Error bars show SEM.

The online version of this article includes the following figure supplement(s) for figure 2:

**Figure supplement 1.** Identification of HpBARI.
**Figure supplement 2.** HpBARI DNA, protein and genomic sequence.
**Figure supplement 3.** C-terminus tagged HpBARI shows no suppressive activity.

## HpBARI suppresses *Alternaria*-induced type two immune responses in vivo

When recombinant HpBARI was co-administered with *Alternaria* allergen, it suppressed *Alternaria*-induced BAL and lung eosinophilia, ILC2 activation and soluble type 2 cytokines 24 hr after allergen administration (*Figure 3A–D* and *Figure 3—figure supplement 1*), at a timepoint where responses are independent of the adaptive immune response (*Bartemes et al., 2012*; *Kouzaki et al., 2011*). This activity was specific to the HpBARI protein, as suppressive effects could not be replicated by administration of *H. polygyrus* Acetylcholinesterase (HpAChE) - a control protein secreted by *H. polygyrus* adult worms (Accession No. JF439067) (*Hewitson et al., 2011*), expressed in the same system as HpBARI (*Figure 3—figure supplements 1* and *2*). Although many of the suppressive effects of N-terminal tagged HpBARI (HpBARI_N), such as suppression of BAL and lung eosinophilia, and ILC2 IL-5 production, were not replicated with less active HpBARI_C (*Figure 3—figure supplement 1A–C* and *Figure 2—figure supplement 3*), trends for suppression by HpBARI_C reached significance for ILC2 IL-13 production and ST2 detection (*Figure 3—figure supplement 1D–G*), indicating this construct retained some activity, albeit much abrogated.

Furthermore, we utilized an *Alternaria*-OVA model of asthma in which *Alternaria* allergen is used to induce a Th2 immune response against OVA protein in the lungs, which can be recalled by later administration of OVA protein alone (*Osbourn et al., 2017*). HpBARI administered at day 0 was able to suppress BAL and lung eosinophilia, and ILC2 and CD4+ T cell IL-13 production, although in the case of ILC2 IL-13 production this did not reach statistical significance (*Figure 4A–D*).

## HpBARI suppresses ST2 detection, but not ST2 transcription

ST2 flow cytometric staining was assessed on lung cells 24 hr after *Alternaria* administration in the presence or absence of HpBARI. HpBARI administration, alone or co-administered with *Alternaria* allergen, resulted in significant abrogation of the ST2 signal to far below baseline levels in the PBS group, on total CD45+ lung cells (*Figure 5A*) or gated ILC2s (*Figure 5B*), similarly to that with HES administration (*Figure 1A–B*). However, when ST2 transcription was assessed by qPCR, no suppression below baseline levels of ST2 transcription was seen with HpBARI administration (*Figure 5C*). We hypothesized that HpBARI was acting directly on the surface ST2 protein, abrogating its detection by flow cytometry.

## The in vivo suppressive effects of HpBARI are persistent

We further investigated HpBARI's effects by administering a biotinylated construct of HpBARI to mice, over a timecourse prior to *Alternaria* administration. We found that HpBARI suppressed

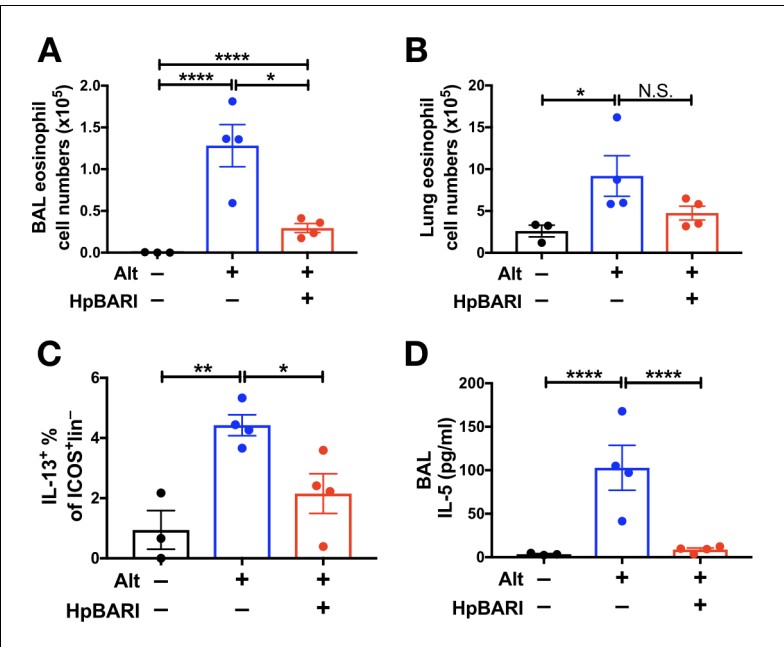

**Figure 3.** HpBARI suppresses *Alternaria*-induced innate immune responses. *Alternaria* allergen (10 μg) and HpBARI (10 μg) were intranasally administered and mice culled 24 hr later. Eosinophils were enumerated in the BAL (A) and lung (B), and intracellular cytokine staining carried out for IL-13 in ICOS$^+$lineage$^-$CD45$^+$ lung ILC2s (C). IL-5 was measured in cell-free BAL supernatants (D). Data representative of 2 repeat experiments, each with 3–4 mice per group. Error bars show SEM.
The online version of this article includes the following figure supplement(s) for figure 3:

**Figure supplement 1.** C-terminus tagged HpBARI shows abrogated suppressive activity, while no suppression is seen with control protein administration.
**Figure supplement 2.** Gating strategy for lung ILC2s.

eosinophilia 24 hr after *Alternaria* allergen administration, even when a single dose of HpBARI was given 24 hr prior to allergen (*Figure 5D*), while lung ILC2 ST2 suppression by HpBARI had similar kinetics (*Figure 5E* and *Figure 5—figure supplement 1D*). Using avidin-PE staining of biotinylated HpBARI, we found that HpBARI could be detected on the surface of lung ILC2s (but not control ST2-negative lung cells) up to 7 days after administration, although this signal only reached statistical significance when HpBARI was co-administered with *Alternaria* allergen (*Figure 5—figure supplement 1*). These data support the hypothesis that HpBARI binds directly to ST2 on the surface of cells, and persists in vivo for at least 48 hr.

## ST2 is systemically suppressed during *H. polygyrus* infection

In order to assess the potential role of HpBARI during *H. polygyrus* infection, mice were orally infected, and tissues collected 7 days later (when the parasite is resident in the subserosal layer of the duodenum). Although little to no detectable ST2 staining was found on ILC2s of the mesenteric lymph node (*Figure 6—figure supplement 1B–D*) (consistent with recent findings [*Ricardo-Gonzalez et al., 2018*; *Schneider et al., 2019*]), strong suppression of ST2 staining was found in the peritoneal lavage and lung ILC2 populations (*Figure 6A–D*, and *Figure 6—figure supplement 1A–C*). Strikingly, a highly granular lineage-negative population in the peritoneal lavage (most likely mast cells [*Enoksson et al., 2013*]) expressed very high levels of ST2 in naïve mice, which was strongly suppressed during *H. polygyrus* infection (*Figure 6E–F*, and *Figure 6—figure supplement 1D*).

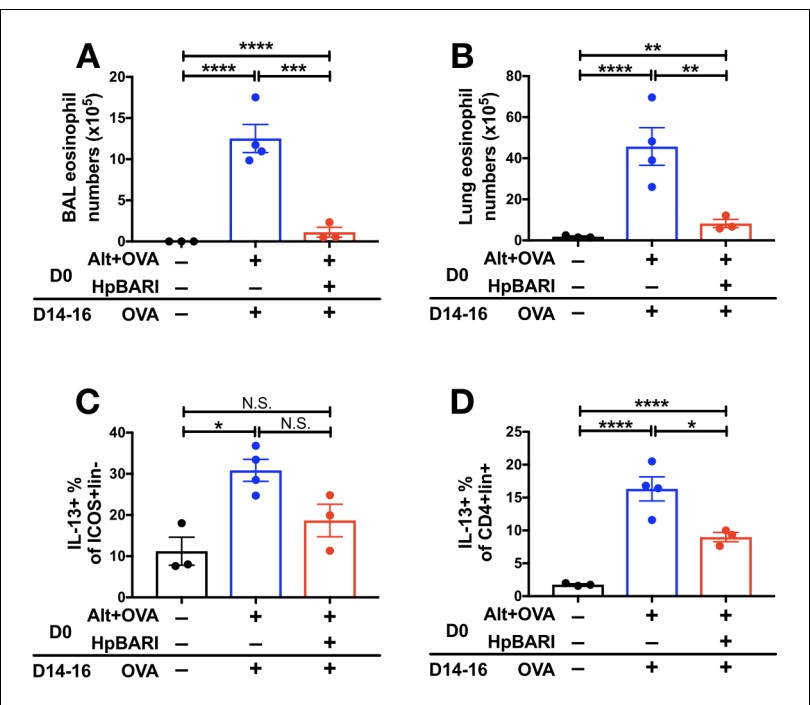

**Figure 4.** HpBARI suppresses *Alternaria*-induced adaptive immune responses. *Alternaria* allergen (10 μg) and OVA protein (20 μg) were coadministered in the presence or absence of HpBARI (10 μg) at day 0 (D0), and the type two immune response recalled two weeks later by three daily administrations of OVA protein, on days 14–16 (D14-16). Eosinophils were enumerated in the BAL (A) and lung (B), and intracellular cytokine staining carried out for IL-13 in ICOS⁺lineage⁻CD45⁺ lung ILC2s (C) and CD4⁺lineage⁺CD45⁺ Th cells (D). Data representative of 2 repeat experiments, each with 3–4 mice per group. Error bars show SEM.

## HpBARI binds to murine ST2, blocking IL-33 ligation

To test the hypothesis that HpBARI was acting directly on the IL-33 receptor complex, a solid-phase ELISA assay was used to assess binding of ST2-Fc (soluble form of ST2 with an Fc tag) or IL-33-TRAP (a fusion protein of ST2 and IL-1RAcP with an Fc tag) (*Holgado et al., 2019*) to plate-bound HpBARI. HpBARI bound mouse ST2 or IL-33-TRAP in a dose-dependent manner, while control plates coated with HpARI (expressed in the same system with the same tags) did not show any binding to these constructs (*Figure 7A*). These data were confirmed using an immunoprecipitation approach, conjugating ST2-Fc and IL-33-TRAP to protein G beads. Confirming the ELISA data, HpBARI was depleted from solution and immunoprecipitated by ST2 or IL-33-TRAP (*Figure 7B*).

To confirm the specificity of HpBARI binding to ST2, biotinylated HpBARI was incubated for 20 min at 4°C with bone marrow-derived ILC2s from either wild-type or ST2-deficient mice. HpBARI bound strongly to wild-type ILC2s, while no binding of biotinylated HpARI to wild-type ILC2s, or biotinylated HpBARI to ST2-deficient ILC2s could be seen (*Figure 7C*). Pre-incubation of ILC2s with either HES or unlabeled HpBARI ablated binding of biotinylated HpBARI, while maintaining suppression of ST2 (*Figure 7D*, and *Figure 7—figure supplement 1A*), further supporting the specificity of this interaction, and showing that HpBARI in HES was responsible for the suppression of ST2 detection. In order to assess whether HpBARI suppression was specific to the antibody used for ST2 detection, we compared ST2 detection by clone RM-ST2-2 (ThermoFisher) to clone DIH9 (Biolegend), but found that HpBARI ablated ST2 detection with both antibodies (*Figure 7—figure supplement 1B*).

Finally, surface plasmon resonance (SPR) was used to assess affinity of binding of HpBARI to the mouse IL-33 receptor. Protein G-coated chips were used to create a stable surface of ST2-Fc and IL-33-TRAP, and the binding of HpBARI was assessed. HpBARI showed high affinity for mouse ST2 ($k_+$=4 μM⁻¹s⁻¹, $k_-$=1.1×10⁻³ s⁻¹, $K_d$ = 0.26 nM) and IL-33-TRAP ($k_+$=3.9 μM⁻¹s⁻¹, $k_-$=6.5×10⁻⁴ s⁻¹, $K_d$ = 0.17 nM) (*Figure 7E*). When C-terminal-tagged HpBARI was tested in the SPR assay, although

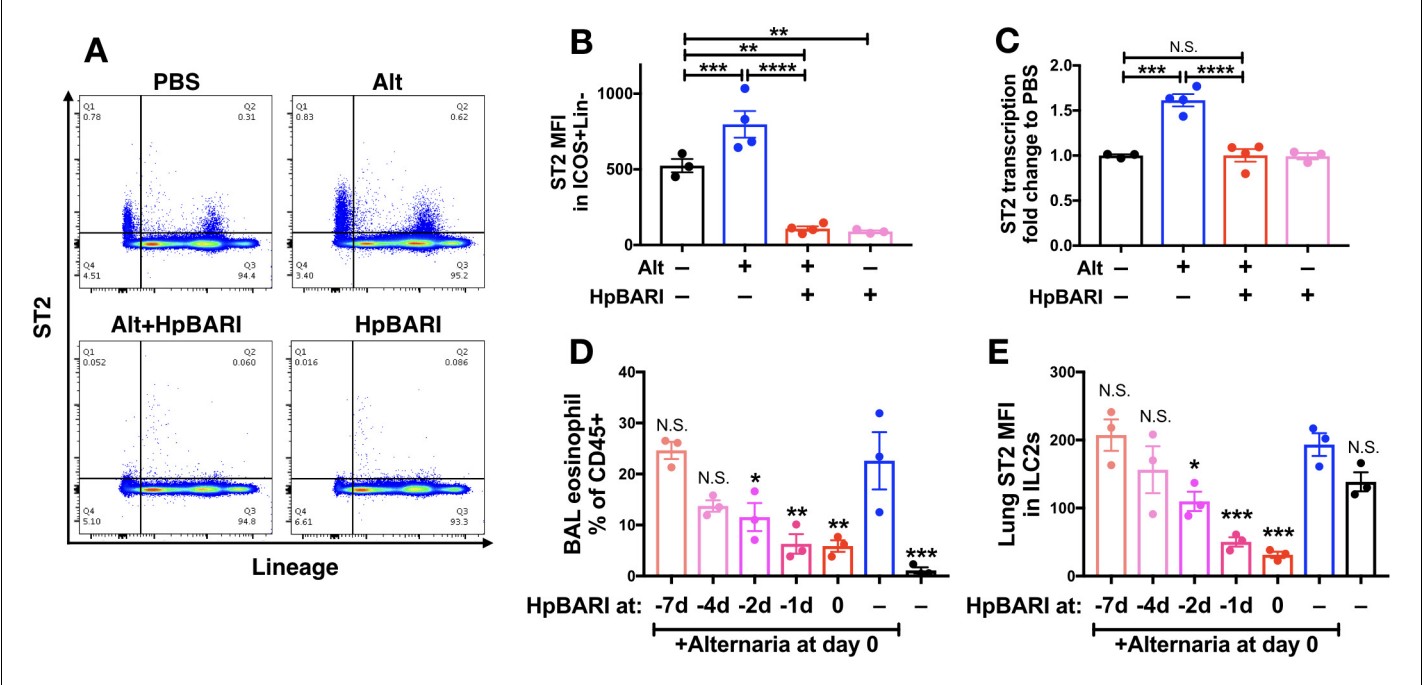

**Figure 5.** HpBARI administration results in persistent suppression of ST2 in vivo. (A–C) *Alternaria* allergen and HpBARI was intranasally administered and mice culled 24 hr later, as in *Figure 3A–D*. Representative ST2 versus lineage bivariate plots (gated on live CD45⁺ cells) are shown in (A). ST2 geometric mean fluorescence intensity on the surface of ICOS⁺lineage⁻CD45⁺ ILC2s was measured by flow cytometry (B). ST2 transcript was measured by qPCR in lung homogenates (C). PBS, Alt and Alt+HpBARI groups are representative of 2 repeat experiments, each with 3–4 mice per group; HpBARI alone group from a single experiment, n = 3. (D–E) Biotinylated HpBARI was intranasally administered 7 days, 4 days, 2 days or 1 day before *Alternaria* allergen, or co-administered (co-ad) with it. 24 hr after *Alternaria* allergen administration, BAL eosinophil (Siglecf⁺CD11c⁻) proportions of CD45⁺ cells (D), and mean fluorescence intensity of ST2 (E) on lung ILC2s (ICOS⁺Lineage⁻CD45⁺) were calculated. Representative results from two repeat experiments, each with three mice per group. Error bars show SEM.

The online version of this article includes the following figure supplement(s) for figure 5:

**Figure supplement 1.** HpBARI is detectable on ILC2 in vivo.

it could still bind to IL-33-TRAP the construct's off-rate was found to be ~4 fold faster than the N-terminal tagged construct (*Figure 7—figure supplement 2*), giving some explanation for its abrogated function (*Figure 2—figure supplement 3* and *Figure 3—figure supplement 1*).

To assess whether the binding of HpBARI to the IL-33 receptor inhibited IL-33 ligation, protein G microbeads were coated with ST2-Fc or IL-33-TRAP, and HpBARI was allowed to bind. IL-33 was then added to the beads and precipitation of IL-33 assessed by western blot, showing that HpBARI could prevent the interaction of IL-33 with its receptor (*Figure 7F*). This interaction was partially abrogated by heat-treatment of HpBARI. The inhibition of IL-33-ST2 interaction by HpBARI was confirmed by surface plasmon resonance: mouse IL-33-TRAP was used to coat a protein G chip, HpBARI was bound, and IL-33 binding assessed. IL-33 gave a robust response on binding to IL-33-TRAP, which was ablated on preincubation of HpBARI (*Figure 7—figure supplement 3*). Thus, HpBARI binds ST2, inhibits binding of IL-33 to its receptor, and ablates IL-33-mediated responses.

## HpBARI_Hom2 binds to human ST2, blocking human IL-33 responses

HpBARI was tested for binding to human ST2 and found to show far lower binding to the human than to the mouse form of the IL-33 receptor (*Figure 8A*). However, homology searching via Wormbase Parasite (*Howe et al., 2017*) identified the *H. polygyrus* gene HPOL_0001228401 encoding a protein with 58% identity to HpBARI which we named HpBARI_Hom2 (*Figure 8—figure supplement 1A*). HpBARI_Hom2 has a similar enough structure to HpBARI that it can be detected on a western blot probed with anti-HpBARI polyclonal rat IgG, which also detects a faint band in HES, indicating

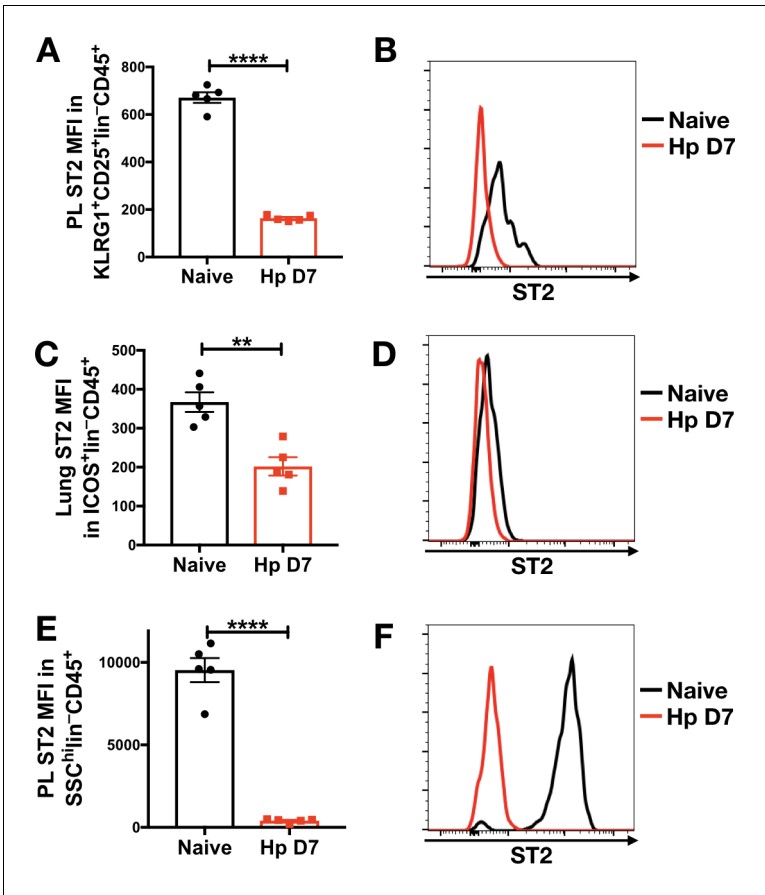

**Figure 6.** ST2 suppression in the peritoneal lavage and lung during *Heligmosomoides polygyrus* infection. Mice were infected with 200 *H. polygyrus* L3 larvae, and at day seven post-infection peritoneal lavage (PL) and lung cells were stained for flow cytometry of ST2 on peritoneal lavage KLRG1$^+$CD25$^+$lineage$^-$CD45$^+$ ILC2s (A, B), lung ICOS$^+$lineage$^-$CD45$^+$ ILC2s (C, D), or peritoneal lavage SSC$^{hi}$lineage$^-$CD45$^+$ cells (E, F). Geometric mean fluorescence intensity of ST2 is shown in A, C and E, representative ST2 histograms on gated cells is shown in B, D and F. Data from a single experiment with five mice per group. Error bars show SEM.

The online version of this article includes the following figure supplement(s) for figure 6:

**Figure supplement 1.** Gating strategy for mesenteric lymph node and peritoneal lavage cells from *Heligmosomoides polygyrus*-infected mice.

the presence of these proteins in endogenous *H. polygyrus* secretions (*Figure 8—figure supplement 1B*).

When this protein was expressed, it was found to show far greater binding to human ST2 (*Figure 8A*) than HpBARI. SPR experiments confirmed that HpBARI_Hom2 bound more strongly to human ST2 (k$_+$=4.8 µM$^{-1}$s$^{-1}$, k$_-$=2.98×10$^{-4}$ s$^{-1}$, K$_d$ = 0.062 nM) than to mouse ST2 (k$_+$=7.18 µM$^{-1}$s$^{-1}$, k$_-$=9.56×10$^{-4}$ s$^{-1}$, K$_d$ = 0.138 nM) (*Figure 8B–C*; *Figure 8—figure supplement 2*).

As with HpBARI and mouse ST2, HpBARI_Hom2 blocked interaction of human IL-33 with human ST2. In a SPR assay, the signal seen when human IL-33 was allowed to bind to a human ST2-coated chip was ablated when HpBARI_Hom2 was injected first (*Figure 8—figure supplement 2B*). Furthermore, when an oxidation-resistant form of human IL-33 (*Cohen et al., 2015*) was coated onto an ELISA plate and probed with human ST2, HpBARI_Hom2 showed a significantly increased ability to block human IL-33-ST2 interaction compared to HpBARI (*Figure 8D*). Finally, in order to test the effects of HpBARI_Hom2 on human cells, we utilized a robust in vitro IL-33 bioassay where PBMCs are stimulated with IL-12 and IL-33, which synergise to induce IFN-γ production from NK cells (*Figure 8—figure supplement 3*; *Smithgall et al., 2008*). HpBARI_Hom2 blocked IL-33-induced IFN-γ production from human PBMCs (average IC50 of 16 nM or 480 ng/ml, measured by ELISA), while

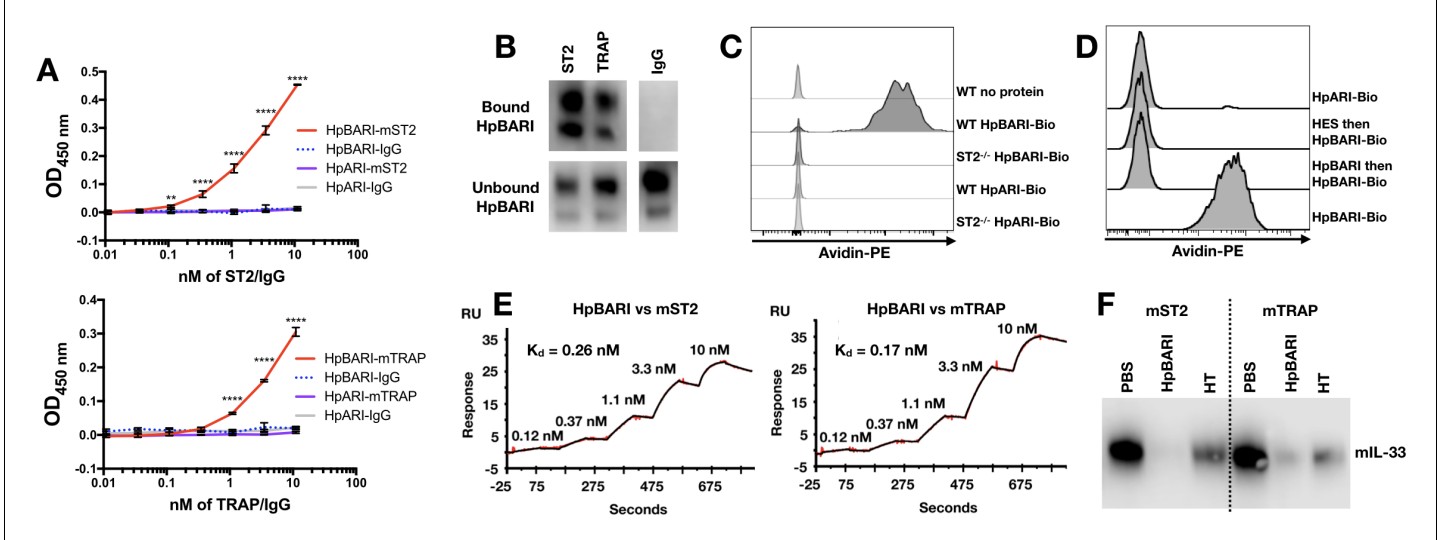

**Figure 7.** HpBARI binds to murine ST2, blocking IL-33 ligation. (A) HpBARI or HpARI (1 µg/ml each) were coated onto wells of an ELISA plate, followed by addition of mouse ST2-Fc (ST2) (top panel) or mouse IL-33 TRAP (bottom panel) or IgG controls, followed by detection using anti-human IgG-HRP. Representative of 3 repeat experiments. Two-way ANOVA comparing HpBARI-mST2 to HpARI-mST2. Error bars show SEM. (B) Mouse ST2-Fc, mouse IL-33 TRAP or IgG were bound to protein G-coated beads, and used to immunoprecipitate HpBARI. Anti-myc western blots shown. Representative of 2 repeat experiments. (C–D) Bone marrow cells of wild type (C–D) or ST2-deficient mice (C) were cultured for 5 days with IL-2, IL-7 and IL-25 (all 10 ng/ml), then incubated with biotinylated HpARI or HpBARI (0.1 µg/ml) for 20 min at 4°C, followed by detection of biotin with avidin-PE. Representative plots gated on ICOS+lineage−CD45+ ILC2. In (D), where indicated samples were treated with HES (10 µg/ml) or unlabelled HpBARI (10 µg/ml) for 20 min prior to HpBARI staining (D). Representative of at least two repeat experiments. (E) Surface plasmon resonance of HpBARI binding to chip-coated mouse ST2 (left panel) or mouse IL-33 TRAP (right panel). (F) Anti-IL-33 western blot, after immunoprecipitation of mouse IL-33 with mouse ST2-Fc or mouse IL-33 TRAP-Fc on protein G dynalbeads, in the presence of HpBARI or heat-treated HpBARI (HT). Representative of 2 repeat experiments. The online version of this article includes the following figure supplement(s) for figure 7:

**Figure supplement 1.** Gating strategy for bone marrow ILC2s.
**Figure supplement 2.** C-terminus tagged HpBARI has decreased affinity for the IL-33 receptor.
**Figure supplement 3.** HpBARI blocks mouse IL-33-TRAP/IL-33 interaction.

HpBARI had far lower activity (IC50 of >1000 nM (*Figure 8E*)). Therefore, HpBARI_Hom2 is a high-affinity binder of human ST2, and a functional suppressor of human IL-33 responses.

## Discussion

IL-33 receptor activation is a potent inducer of type two immune responses which mediate the ejection of helminth parasites (as well as inducing allergic disease), and so the IL-33 pathway represents a central target for parasite immunomodulation. Here, we identify HpBARI, a protein secreted by the intestinal nematode *H. polygyrus*, which binds and blocks ST2, abrogating IL-33 responses. HpBARI prevents IL-33-dependent responses to intranasal *Alternaria* allergen, and abrogated detection (but not transcription) of ST2 in lung cells. HpBARI binds strongly to ST2, preventing IL-33 ligation of the receptor. HpBARI is present in HES, and during *H. polygyrus* infection ST2 detection is abrogated on peritoneal and lung cells, indicating systemic effects of parasite immunomodulation. Furthermore, we identified HpBARI_Hom2, a related protein from *H. polygyrus* which is capable of suppressing human IL-33 responses through the same mechanism.

HpBARI joins the suite of immunomodulatory activities of *H. polygyrus*, many of which are directed against the IL-33 pathway. We previously identified HpARI from *H. polygyrus* products, a protein which binds IL-33 and genomic DNA, preventing the release of the alarmin cytokine by necrotic host cells (*Osbourn et al., 2017*). *H. polygyrus* also releases miRNA-containing extracellular vesicles which downregulate the transcription of ST2 (*Buck et al., 2014*) and an unknown factor which induces IL-1β release, counter-regulating IL-33 responses (*Zaiss et al., 2013*). Therefore, the IL-33 pathway, and its modulation, appears to be of crucial importance to this parasite.

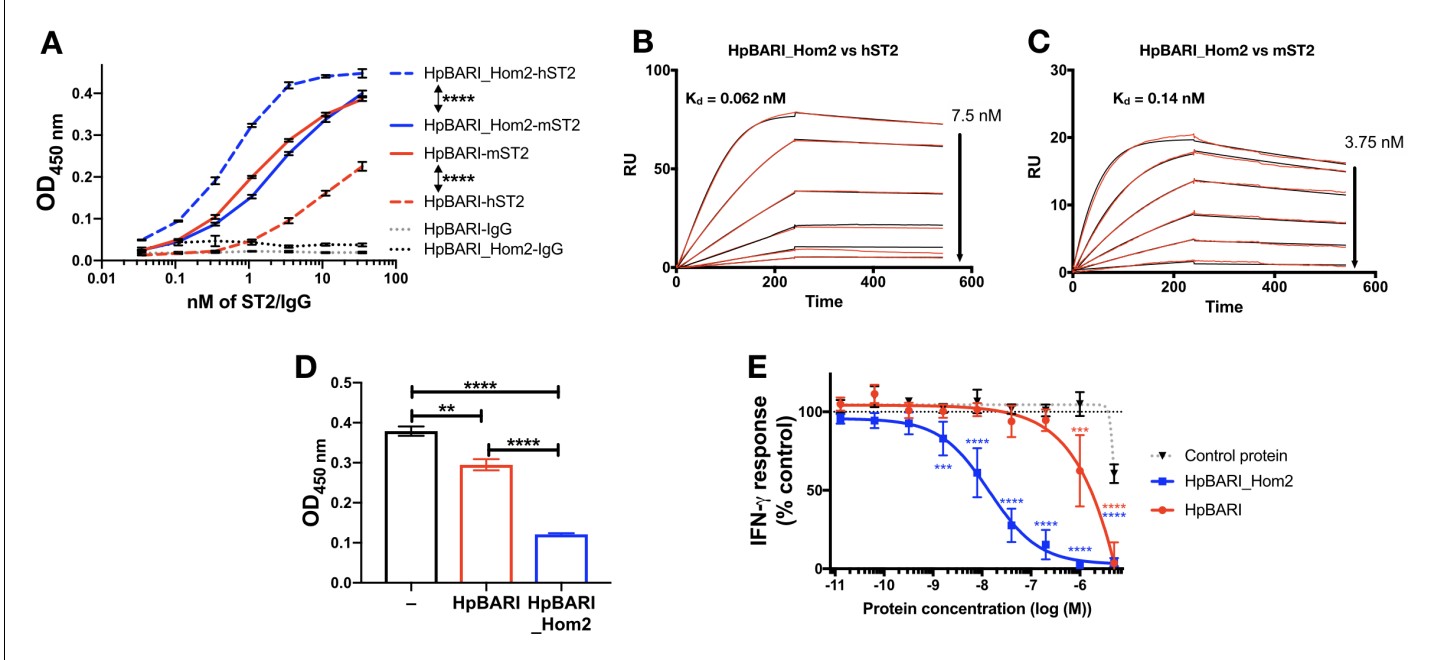

**Figure 8.** HpBARI_Hom2 binds to human ST2, blocking human IL-33 responses. (A) HpBARI or HpBARI_Hom2 (1 μg/ml each) were coated onto wells of an ELISA plate, followed by addition of mouse ST2-Fc (mST2), human ST2-Fc (hST2) or IgG controls, followed by detection using anti-human IgG-HRP. Representative of 2 repeat experiments. Two-way ANOVA comparing hST2 to mST2 binding by HpBARI and HpBARI_Hom2. (B–C) Surface plasmon resonance of HpBARI binding to chip-coated mouse (B) or human ST2-Fc (C). (D) An oxidation-resistant mutant of human IL-33 (1 μg/ml) was coated onto wells of an ELISA plate. Human ST2-Fc was then added to the plate, either alone or after incubation with HpBARI or HpBARI_Hom2, followed by detection of IL-33 binding by ST2-Fc using anti-human IgG-HRP. Representative of 2 repeat experiments. (E) Human PBMCs were stimulated with 5 ng/ml human IL-12, 0.1 ng/ml oxidation resistant human IL-33 and HpBARI, HpBARI_Hom2 or a control protein (HpAChE) for 44 hr. IFN-γ in the supernatant was measured by ELISA with values expressed as % maximal response (IFN-γ release) where 100% is IFN-γ release in response to IL-12 and IL-33 co-stimulation, corrected for background levels in each plate/donor. The graph shows curves pooled from three donors where identical dose-response curves were performed. Two-way ANOVA compares HpBARI and HpBARI_Hom2 to control protein treatment. Error bars show SEM.
The online version of this article includes the following figure supplement(s) for figure 8:

**Figure supplement 1.** Alignment of HpBARI and HpBARI_Hom2 amino acid sequence, and wester blot of HpBARI, HpBARI_Hom2 and HES.
**Figure supplement 2.** HpBARI_Hom2 inhibits human ST2/IL-33 interaction.
**Figure supplement 3.** Stimulation of PBMC IFN-y production by IL-12 and IL-33.

During *H. polygyrus* infection, we found abrogated detection of ST2 on ST2-expressing cells such as ILC2s in the peritoneal lavage and lung. Recent papers have noted that ST2 is expressed at a far lower level on ILC2s in the intestine compared to those from the lung, fat or bone marrow (*Ricardo-Gonzalez et al., 2018*). Therefore it may be that *H. polygyrus* acts on immune cells outside of the intestine, rendering them unresponsive to IL-33, and suppressing their activity prior to their recruitment to the site of infection. Interestingly, extracellular vesicles secreted by *H. polygyrus* have also been shown to reduce pulmonary ST2 expression when delivered directly to the lung (*Buck et al., 2014*), but our data are the first to suggest systemic reduction in ST2 detection during intestinal helminth infection.

Our ELISA, immunoprecipitation, SPR and flow cytometry data show that HpBARI can bind effectively to both membrane-bound and soluble forms of ST2. During *H. polygyrus* infection, HpBARI may therefore bind soluble ST2 in circulation, abrogating the effects of the parasite immunomodulator. Blockade of soluble ST2 (rather than membrane-bound ST2) by anti-ST2 antibodies in graft-versus host disease has been suggested to enhance the effects of IL-33, by removing an endogenous inhibitor of IL-33 (*Zhang et al., 2015*), but it is still unclear whether these effects seen are due to enhancement or blockade of IL-33 signalling (*Scott et al., 2016*). Our data suggest that HpBARI suppresses, rather than enhances, IL-33 responses.

On release, IL-33 is processed by endogenous and exogenous proteases from the full-length protein, stored in the nucleus, to a series of smaller processed forms with increased activity

(*Cayrol et al., 2018*; *Scott et al., 2018*). Full-length IL-33 is presumed to have a lower affinity for ST2 than processed forms, however these measurements have not to date been directly made. HpBARI can outcompete the highest-affinity mature form of IL-33 for binding to ST2, therefore we conclude that HpBARI could abrogate signals from all forms of IL-33.

The advantage of targeting ST2 rather than IL-33 is that ST2 is constitutively expressed on the surface of many cell types, and therefore could be blocked prior to damage and IL-33 release. IL-33, while constitutively expressed in the nucleus of epithelial cells, is released transiently on damage, and has a very short-lived active form (*Cohen et al., 2015*). Therefore, HpBARI could prevent the responsiveness of ST2-positive cells to IL-33 before the cytokine has been released.

Intriguingly, HpBARI and HpBARI_Hom2 are CCP domain proteins, sharing sequence motifs with the previously-identified *H. polygyrus* immunomodulators HpARI and the Hp-TGM family (*Johnston et al., 2017*; *Osbourn et al., 2017*; *Smyth et al., 2018*). Therefore, the identification of HpBARI as a suppressor of ST2 signalling cements *H. polygyrus* CCP domain-containing proteins as an immunomodulatory group of secreted proteins. The *H. polygyrus* genome contains >80 CCP domain-containing proteins (*Maizels et al., 2018*), many of which can be detected in the parasite's secretions (*Hewitson et al., 2013*). We now extend the functions of the CCP domain-containing family: HpBARI is a cytokine receptor antagonist, in contrast to Hp-TGM, a suppressive cytokine receptor agonist, and HpARI, a cytokine antagonist. The remainder of this large secreted family of proteins remains to be characterized, and we hypothesise that other modulators of cytokine/cytokine receptor interactions are contained therein.

As parasite genomes and secretions are becoming better understood, the interactions between parasite proteins and the immune system is becoming further characterized. Recently, an IL-13-binding protein from the intestinal parasite *Trichuris muris* was identified (*Bancroft et al., 2019*), indicating that direct binding interactions with molecules of the host immune system are not unique to *H. polygyrus*.

We found that a homologue of HpBARI is also expressed by *H. polygyrus*, and that this homologue (HpBARI_Hom2) had considerably higher affinity for human ST2. As *H. polygyrus* is not human-infective, it is unlikely that HpBARI_Hom2 evolved to bind human ST2, however this selectivity of binding may be due to the host range of the parasite. Strains of *H. polygyrus* infect wood mice (*Apodemus sylvaticus*), groove-toothed harvest mice (*Reithrodontomys* spp.) and reed voles (*Microtus* spp) as well as *Mus musculus* (*Forrester, 1971*; *Kim et al., 2015*), host species which have a common ancestor 15–20 million years ago (*Steppan and Schenk, 2017*). Thus *H. polygyrus* may have developed subfamilies of its immunomodulators, which have been selected for peak efficacy in each host species.

In conclusion, we here define HpBARI and HpBARI_Hom2, novel immunomodulatory proteins secreted by the murine intestinal nematode *H. polygyrus*. HpBARI and HpBARI_Hom2 bind directly to ST2, blocking interaction of IL-33 with its receptor, and preventing IL-33-mediated responses. We hypothesise that HpBARI inhibits the initiation of type two immune responses during damage-induced IL-33 release in nematode infection. This study extends the knowledge of parasite immunomodulation, and the interaction between parasite proteins and allergic sensitization. By learning how model parasitic nematodes modulate the host immune response, we may be able to design better vaccines targeted at removing parasite-derived immunosuppression, allowing host ejection of the parasite (*McNeilly and Nisbet, 2014*). Furthermore, by learning from these 'masters of regulation' (*Maizels et al., 2004*), we may be able to develop new treatments for human allergic disease.

## Materials and methods

### Key resources table

| Reagent type (species) or resource | Designation | Source or reference | Identifiers | Additional information |
|---|---|---|---|---|
| Strain, strain background, *Mus musculus* | ST2-deficient; *Il1rl1$^{-/-}$* | (*Mangan et al., 2007*) | | BALB/c background |

*Continued on next page*

*Continued*

| Reagent type (species) or resource | Designation | Source or reference | Identifiers | Additional information |
|---|---|---|---|---|
| Cell line (*Homo sapiens*) | Expi293F | ThermoFisher Scientific | Cat No: A14527 | |
| Recombinant DNA reagent | pSecTAG2A vector | ThermoFisher Scientific | Cat No: V90020 | |
| Other | *Alternaria alternata* allergen extract | Greer | Cat No: XPM1D3A25 | |
| Peptide, recombinant protein | Mouse IL-2 | Biolegend | Cat No: 575402 | |
| Peptide, recombinant protein | Mouse IL-7 | Biolegend | Cat No: 577802 | |
| Peptide, recombinant protein | Mouse IL-33 | Biolegend | Cat No: 580502 | |
| Peptide, recombinant protein | Mouse IL-25 | Biolegend | Cat No: 587302 | |
| Peptide, recombinant protein | Human IL-33 | Biolegend | Cat No: 581802 | |
| Peptide, recombinant protein | Mouse ST2-Fc chimera | Biolegend | Cat No: 764902 | |
| Peptide, recombinant protein | Human ST2-Fc chimera | Biolegend | Cat No: 557904 | |
| Antibody | Anti-CD3-FITC (Armenian hamster monoclonal) | Biolegend | Cat No: 100306 RRID:AB_312671 Clone: 145–2 C11 | (1:200) |
| Antibody | Anti-CD5-FITC (Rat monoclonal) | Biolegend | Cat No: 100606 RRID:AB_312735 Clone: 53–7.3 | (1:200) |
| Antibody | Anti-CD11b-FITC (Rat monoclonal) | Biolegend | Cat No: 101224 RRID:AB_755986 Clone: M1/70 | (1:200) |
| Antibody | Anti-CD19-FITC (Rat monoclonal) | Biolegend | Cat No: 115506 RRID:AB_313641 Clone: 6D5 | (1:200) |
| Antibody | Anti-CD49b-FITC (Rat monoclonal) | ThermoFisher Scientific | Cat No: 11-5971-85 RRID:AB_465327 Clone: DX5 | (1:200) |
| Antibody | Anti-GR1-FITC (Rat monoclonal) | Biolegend | Cat No: 108406 RRID:AB_313371 Clone: RB6-8C5 | (1:200) |
| Antibody | Anti-CD45-AlexaFluor700 (Rat monoclonal) | Biolegend | Cat No: 103128 RRID:AB_493715 Clone: 30-F11 | (1:200) |
| Antibody | Anti-CD45-Pacific Blue (Rat monoclonal) | Biolegend | Cat No: 103126 RRID:AB_493535 Clone: 30-F11 | (1:200) |
| Antibody | Anti-ICOS-PCP-Cy5.5 (Armenian hamster monoclonal) | Biolegend | Cat No: 313518 RRID:AB_10641280 Clone: C398.4A | (1:100) |
| Antibody | Anti-CD4-PE-Dazzle (Rat monoclonal) | Biolegend | Cat No: 100566 RRID:AB_2563685 Clone: RM4-5 | |

*Continued on next page*

*Continued*

| Reagent type (species) or resource | Designation | Source or reference | Identifiers | Additional information |
|---|---|---|---|---|
| Antibody | Anti-ST2-APC (Rat monoclonal) | Thermo Fisher Scientific | Cat No: 17-9335-82 RRID:AB_2573301 Clone: RMST2-2 | (1:100) Used throughout manuscript. |
| Antibody | Anti-ST2-APC (Rat monoclonal) | Biolegend | Cat No: 145306 RRID:AB_2561917 Clone: DIH9 | (1:100) Used in *Figure 7—figure supplement 1* only |
| Antibody | Anti-CD11c-Alexa Fluor647 (Armenian hamster monoclonal) | Biolegend | Cat No: 117312 RRID:AB_389328 Clone: N418 | (1:200) |
| Antibody | Anti-SiglecF-PE (Rat monoclonal) | Miltenyi | Cat No: 130-102-274 RRID:AB_2653451 Clone: ES22-10D8 | (1:50) |
| Antibody | Anti-KLRG1-PCP-Cy5.5 (Syrian hamster monoclonal | Biolegend | Cat No: 138417 RRID:AB_2563015 Clone: 2F1 | (1:100) |
| Antibody | Anti-IL-5-PE (Rat monoclonal) | Biolegend | Cat No: 504304 RRID:AB_315328 Clone: TRFK5 | (1:200) |
| Antibody | Anti-IL-13-PE-Cy7 (Rat monoclonal) | ThermoFisher Scientific | Cat No: 25-7133-82 RRID:AB_2573530 Clone eBio13A | (1:200) |
| Peptide, recombinant protein | Streptavidin-PE | Biolegend | Cat No: 405203 | |
| Kit | Live/dead fixable blue | Thermo Fisher Scientific | Cat No: L34962 | |

## Cell lines and reagents

Expi293F cells were cultured as described in the manufacturer guidelines. Cells were grown in Expi293F Expression Medium (Gibco) at 37°C 8% $CO_2$, shaking at 125 rpm. Expi293F cells were purchased from ThermoFisher Scientific, tested and found to be mycoplasma free.

*Alternaria alternata* extract (Greer XPM1D3A25) was resuspended in PBS and concentration assessed by Pierce BCA assay (Thermo Fisher Scientific).

Mouse and human IL-33-TRAP-Fc, and human oxidation-resistant IL-33 (*Cohen et al., 2015*) were kindly provided by E. Suzanne Cohen (AstraZeneca, Cambridge, UK).

## Animals

BALB/cAnNCrl, C57BL/6JCrl mice were purchased from Charles River, UK. ST2-deficient mice (BALB/c background) were bred in-house. All mice were accommodated and procedures performed under UK Home Office licenses with institutional oversight performed by qualified veterinarians.

## Murine lung single cell suspension

Murine lungs were digested in 2 U/ml of Liberase TL (Roche, Burgess Hill, UK) and 80 U/ml DNase (Life technologies, Paisley, UK) at 37°C with agitation for 35 min. Digested tissue was passed through a 70 μm strainer and red blood cells lysed using Red Blood Cells Lysis Buffer (Sigma). Live cells were counted using a haemocytometer and dead cells excluded using trypan blue.

## Bone marrow isolation

Tibias and femurs were obtained from euthanised mice. Bones were placed in 70% ethanol for 5 min and washed with PBS. Bone marrow was flushed using a syringe containing RPMI 1640 (Gibco) and passed through a 70 μm strainer. Red blood cells were lysed using red blood cell lysis buffer (Sigma).

## HES preparation

HES products were prepared as previously described (*Johnston et al., 2015*). *H. polygyrus* adult parasites were retrieved from the intestines of mice infected 14 days previously, and cultured at 37°C, 5% $CO_2$ in RPMI 1640 containing 1.2% glucose, 5 U/ml penicillin, 5 µg/ml streptomycin, 2 mM L-glutamine and 1% gentamycin. After 24 hr of culture, supernatant was exchanged for fresh medium (first 24 hr supernatant discarded), and then medium was collected and exchanged at least twice per week for 21 days. Collected supernatant was concentrated over a 3 kDa Amicon (Merck) stirred cell ultrafiltration device pressurised under nitrogen, buffer exchanged into PBS, and then protein concentration measured at A280 nm.

HES fractions and proteomic data were obtained from samples described in previous studies (*Johnston et al., 2017*; *Osbourn et al., 2017*). Briefly, a Superdex 200 10/300 GL column was used for size fractionation, to give 25 × 1 ml fractions from approximately 1 mg of starting material. A MonoQ 5/50 GL 1 ml column was used for charge fractionation, with a 40 column-volume gradient from 20 mM TrisHCl pH 8 to a maximum of 30% elution buffer (20 mM TrisHCl + 1 M NaCl pH 8) to obtain 1 ml charge fractions. All HES size and charge fractions were trypsinised and analysed on an on-line LC MS/MS system consisting of a capillary-pump Agilent 1200 HPLC (Agilent, UK) and an Orbitrap XL mass spectrometer (Thermo Scientific), and analysed using Mascot (v2.4, Matrix Science).

Each HES fraction was added to suspensions of murine lung cells (1 µl of fraction in a 200 µl culture volume) and cultured for 24 hr prior to analysis of surface ST2 by flow cytometry. Fractions showing significant suppression of ST2 detection were correlated with abundance of each protein (estimated by emPAI (exponentially modified protein abundance index)) to identify candidates.

## Helminth-derived recombinant protein and protein biotinylation

The HpBARI, HpBARI_Hom2 and HpAChE (Accession No. JF439067) (*Hewitson et al., 2011*) genes were codon-optimised for *Homo sapiens* and gene synthesised (GeneArt, Thermo Fisher). HpBARI was synthesised with either sequences encoding 5' 6-His and c-myc tags, a TEV cleavage site and a 3' stop codon (HpBARI_N), or 3' tags (HpBARI_C). HpBARI_Hom2 was gene synthesised with N-terminal tags, while HpACHE was synthesised with C-terminal tags. Each construct was cloned into a pSecTAG2A expression vector (Thermo Fisher) and transfected into Expi293F cells using manufacturer's instructions for the Expi293F Expression System (Gibco). Protein in supernatants was collected 7 days after transfection, and purified using a HisTrap excel column (GE Healthcare) and 500 mM imidazole used to elute the bound protein. Fractions containing protein of interest were pooled, dialysed to PBS, and repurified on a HiTRAP chelating HP column (GE Healthcare) charged with 0.1 M $NiSO_4$. Elution was performed using an imidazole gradient and fractions positive for protein pooled, dialysed to PBS and filter-sterilised. Protein concentration was measured via A280 nM spectrometry (using calculated extinction coefficient).

Biotinylation was performed using EZ-Link Sulfo-NHS-LC-LC-Biotin (Thermo Fisher scientific) according to the manufacturer's instructions.

## In vitro bone marrow cultures

Bone marrow cells ($5 \times 10^5$ cells per well) were plated in a round bottom 96-well plate and co-cultured with IL-2, IL-7, IL-33 at 10 ng/ml (Biolegend), and recombinant parasite-derived proteins, at 37°C, 5% $CO_2$ for 5 days. Supernatants were collected and analysed by IL-5, IL-6 and IL-13 ELISA according to manufacturer's instructions (Thermo Fisher). Where biotinylated constructs were used to assess binding to bone marrow ILC2s, BALB/c or ST2-deficient bone marrow cells were cultured for 5 days in IL-2, IL-7 and IL-25 at 10 ng/ml (Biolegend) to support ILC2 proliferation, prior to incubation for 20 min at 4°C with HES and recombinant proteins, followed by biotinylated recombinant proteins for a further 20 min, as indicated, and avidin-PE to detect binding.

## *Alternaria* model

*Alternaria* allergen was used as a model of asthma and IL-33 dependent responses as previously described (*McSorley et al., 2014*; *Osbourn et al., 2017*).

*Alternaria* allergen (10 µg) and HpBARI (10 µg) were administered intranasally to mice and mice culled 24 hr after administration. Where indicated, *Alternaria* was administered with OVA (20 µg)

and HpBARI (10 μg). The OVA-specific response was recalled by daily intranasal administration of 20 μg OVA protein on days 14, 15 and 16. To test the effectiveness of HpBARI, the protein (10 μg) was administered intranasally at 7, 4, 2, 1 days prior *Alternaria* (10 μg) administration or co-administrated with the allergen extract. Bronchoalveolar lavage was collected (4 lavages with 0.5 ml ice-cold PBS), followed by lung dissection for tissue digestion and RNA extraction.

## *H. polygyrus* infection

C57BL/6 mice were infected with 200 *H. polygyrus* L3 larvae by oral gavage. Mice were culled 7 days later and peritoneal lavage and lungs harvested. Peritoneal cells were collected by lavaging the peritoneum with 10 ml ice-cold RPMI 1640, and lung single cell suspensions were prepared as described above.

## Flow cytometry staining

For surface staining single cell suspensions were washed in PBS and stained with Fixable Blue Live/Dead stain (Thermo Fisher). Cells were then blocked with anti-mouse CD16/32 antibody (Biolegend) and surface stained. For intracellular cytokine staining, single lung cell suspensions were stimulated with PMA (500 ng/ml), ionomycin (1 μg/ml) and brefeldin A (10 μg/ml) for 4 hr at 37°C, 5% $CO_2$. Stimulated cells were surface stained, and permeabilised with IC permeabilization buffer (eBioscience). Cells were labelled with a cocktail of Lineage markers made of FITC- conjugated antibody specific for CD3, CD5, CD11b, CD19, CD49b and GR1, as well as stains for CD45, ICOS, CD4, ST2, CD11c, and Siglec-F. For intracellular cytokine staining, cells were then stained for IL-5 and IL-13.

## RNA extraction, reverse transcription and qPCR

Lung tissue was placed in RNALater Stabilizing Solution (Thermo Fisher Scientific) and stored at −20°C. Lung tissues were then transferred from RNALater to TRIzol (Thermo Fisher Scientific) and homogenised using 3 mm stainless steel beads (Qiagen) in a TissueLyser II (Qiagen) at 25 Hz for 2 min.

Complementary DNA was made using High-Capacity cDNA Reverse Transcription Kit (Applied Biosystems by Thermo Fisher Scientific). ST2 primers were purchased from Life Technologies and amplification reaction was carried out in a 25 μl volume made up of 1.25 μl of pre-made primer probe mix, 12.5 μl TaqMan mastermix; 7.5 μl H20; 1.25 μl housekeeping primer; 2.5 μl DNA template. Amplification was carried out using StepOne 48-well plate (Applied Biosystems). PCR data were analysed using the $2^{-\Delta\Delta CT}$ method. In brief, relative gene expression between different samples was calculated using the threshold cycles (CTs) generated by StepOne 48-well plate machine and software. ΔCT for each sample was calculated subtracting CT of the housekeeping gene (RPL37) from the CT of the gene of interest. To obtain the relative gene expression between control group and treatment, ΔΔCTs were then obtained subtracting the average of the control group ΔCTs (e.g. PBS) from the ΔCT of the sample. Subsequently, $2^{-\Delta\Delta CT}$ was calculated and plotted in a graph.

## Solid-phase ELISA

Corning Costar 96-well EIA/RIA plate (Fisher Scientific, Thermo Fisher, UK) were coated overnight at 4°C with 1 μg/ml of HpBARI, HpARI or oxidation-resistant IL-33, diluted in 1X Coating buffer (eBioscience) (50 μl/well). Plates were washed three times with ELISA wash buffer (TBS+0.05% Tween 20) and blocked with 150 μl/well ELISA block buffer (PBS 0.5% BSA) for 1 hr at room temperature. Mouse or human ST2-Fc or IL-33-TRAP-Fc, or human IgG were diluted in block buffer starting at 11.1 nM and 10-fold diluted. Constructs were added to the protein-coated plate (50 μl/well) and incubated for 2 hr at room temperature. After incubation, the plate was washed four times with ELISA wash buffer. Anti-human IgG HRP (Invitrogen) diluted 1:3000 in ELISA block buffer (50 μl/well) was incubated for 1 hr at room temperature. Plate was washed four times and 1X TMB substrate was used (50 μl/well), stopping the enzymatic reaction with 2N $H_2SO_4$ (50 μl/well). Absorbance was read at 450 nm.

## Immunoprecipitation

Protein G dynabeads (Thermo Fisher) were coated with mouse ST2-Fc, mouse IL-33-TRAP-Fc or IgG (1 μg). Conjugated beads were washed on a DynaMag-2 magnet with PBS 0.02% Tween 20. These were then used to immunoprecipitate HpBARI and mouse IL-33. Eluted proteins and unbound

supernatants were ran on 4–12% SDS-PAGE gels (Thermo Fisher) under reducing conditions, and transferred to nitrocellulose membranes for western blotting.

## Surface Plasmon Resonance (SPR)

HpBARI SPR was performed at the Edinburgh Protein Production Facilities (EPPF) at the University of Edinburgh. Measurements were performed using single cycle kinetics on a BIAcore T200 instrument (GE Healthcare). Mouse ST2-Fc and mouse TRAP-Fc at 10 nM in 10 mM $NaH_2PO_4$, pH 7.5; 150 mM NaCl; 50 μM EDTA; 0.05% surfactant P20, were immobilised by variable contact time, at 30 μl·min$^{-1}$, on a Protein G sensor chip to ~200 response unit (RU) for mouse ST2-Fc and to ~400 RU for mouse TRAP-Fc. SPR kinetic titration binding experiments were performed at 25°C. Three-fold dilution series of HpBARI (0.12–10 mM), were injected over the sensor surface, in 10 mM $NaH_2PO_4$, pH 7.5; 150 mM NaCl; 50 μM EDTA; 0.05% surfactant P20, at 30 μl.min$^{-1}$ for 90 s followed by a 60 s dissociation phase. A single 10 nM injection for 90 s of HpBARI or IL-33 were performed to study IL-33 binding. The on- ($k_+$) and off-rate ($k_-$) constants and the equilibrium dissociation constant ($K_d$) were calculated by global fitting all three surfaces simultaneously to a 1:1 interaction model, with mass transport considerations, to the double reference corrected sensorgrams, using analysis software (v.2.01, GE Healthcare) provided with the BIAcore T200 instrument. Both interactions were extremely well fit by a simple 1:1 interaction model (Chi$^2$ values of ≤0.1) and showed no evidence of mass transport issues.

Affinity measurements for HpBARI_Hom2 were performed using multicycle kinetics in PBST buffer (PBS + 0.01% Tween20). Approximately 200 RU of human or mouse ST2-Fc were captured on a Protein A/G chip. HpBARI_Hom2 was reconstituted in PBST buffer at multiple two-fold dilutions (7.5 nM to 0.05 nM) and each was injected at 60 μl/minute for 240 s, with a 300 s dissociation time. The sensor surface was regenerated with 10 mM Glycine pH 1.8 between each cycle. Sensorgrams were fitted to a specific one-site binding model to derive on- and off-rates and the $K_d$ values. For competition binding assays, approximately 200 RU of human ST2-Fc was immobilized to Protein A/G chip. HpBARI_Hom2 (15 nM) was injected at a flow rate of 60 μl/min. After 100 s, human IL-33 (15 nM) was flowed over HpBARI_Hom2, held onto the chip surface by immobilized human ST2-Fc. The sensor surface was regenerated as described above.

## PBMC assay

Human PBMCs were prepared from healthy donors under informed consent, with ethical approval under Research Tissue Bank ethics number RTB 16/EE/0334. PBMCs were isolated from leukocyte cones using a polysucrose (Ficoll) gradient and maintained in RPMI 1640 (supplemented with 10% v/v heat-inactivated FBS, 1% v/v Penicillin/Streptomycin) at 37°C in a 5% $CO_2$, humidified atmosphere. PBMCs were stimulated with 5 ng/ml human IL-12, 0.1 ng/ml oxidation resistant human IL-33 (human IL-33$^{112-270}$, C208S, C227S, C232S and C259S) and HpBARI, HpBARI_Hom2 or control protein (HpAChE, Hp_I12803_IG04747_L2174). *Cells* were incubated for 44 hr. Media supernatants were collected and soluble IFN-γ was assessed by ELISA, using anti-human IFN-γ capture antibody (Pharmingen, #551221), biotinylated anti-human IFN-γ detection antibody (Pharmingen, #554550) and DELFIA Europium visualisation (PerkinElmer). Data were analysed using Graphpad Prism software. IC50 values were determined by curve fitting using a four-parameter logistic equation, and a mean value taken of the response of 5 donors. Pooled data from 3 of the 5 donors, where identical titrations of protein were carried out, are shown in *Figure 8E*.

## Statistical analysis

All data were analysed using Prism (Graphpad Software Inc), one-way ANOVA with Dunnet's multiple comparisons post-test was used to compare multiple independent groups, while two-way ANOVA and Tukey's multiple comparison's post-test was used to compare multiple timepoints or concentrations between independent groups. Where necessary, data were log-transformed to give a normal distribution and to equalise variances. Sample size estimation was carried out when the study was being designed, using power calculations to give an 80% power to detect a 50% suppression of responses at p<0.05. ****=p < 0.0001, ***=p < 0.001, **=p < 0.01, *=p < 0.05, N.S. = Not Significant (p>0.05).

## Acknowledgements

This was work was funded by awards to HJM from LONGFONDS | Accelerate as part of the AWWA project, and the Medical Research Council (MR/S000593/1).

## Additional information

### Competing interests

Elizabeth C Hinchy, Josquin A Nys, E Suzanne Cohen: Employee of the AstraZeneca Group and has stock/stock options in AstraZeneca. Henry J McSorley: Author is an inventor on a patent application based on the research of this paper. The other authors declare that no competing interests exist.

### Funding

| Funder | Grant reference number | Author |
|---|---|---|
| Longfonds | Accelerate | The AWWA project | Caroline Chauché<br>Henry J McSorley |
| Medical Research Council | MR/S000593/1 | Francesco Vacca<br>Adefunke Ogunkanbi<br>Henry J McSorley |

The funders had no role in study design, data collection and interpretation, or the decision to submit the work for publication.

### Author contributions

Francesco Vacca, Conceptualization, Formal analysis, Investigation, Writing - original draft, Writing - review and editing; Caroline Chauché, Conceptualization, Formal analysis, Investigation, Writing - review and editing; Abhishek Jamwal, Elizabeth C Hinchy, Martin Wear, Formal analysis, Investigation, Methodology, Writing - review and editing; Graham Heieis, Holly Webster, Adefunke Ogunkanbi, Zala Sekne, William F Gregory, Investigation, Methodology, Writing - review and editing; Georgia Perona-Wright, Conceptualization, Supervision, Investigation, Methodology, Writing - review and editing; Matthew K Higgins, Conceptualization, Resources, Formal analysis, Supervision, Investigation, Methodology, Writing - review and editing; Josquin A Nys, Conceptualization, Resources, Formal analysis, Investigation, Methodology, Writing - review and editing; E Suzanne Cohen, Conceptualization, Resources, Formal analysis, Supervision, Methodology, Writing - review and editing; Henry J McSorley, Conceptualization, Data curation, Supervision, Funding acquisition, Writing - original draft, Project administration, Writing - review and editing

### Author ORCIDs

Francesco Vacca (iD) https://orcid.org/0000-0001-6808-5221
Matthew K Higgins (iD) http://orcid.org/0000-0002-2870-1955
Henry J McSorley (iD) https://orcid.org/0000-0003-1300-7407

### Ethics

Human subjects: Human PBMCs were prepared from healthy donors under informed consent, with ethical approval under Research Tissue Bank ethics number RTB 16/EE/0334.
Animal experimentation: All mice were accommodated and procedures performed under UK Home Office licenses with institutional oversight performed by qualified veterinarians. UK Home Office project license number 70/8733.

### Decision letter and Author response

Decision letter https://doi.org/10.7554/eLife.54017.sa1
Author response https://doi.org/10.7554/eLife.54017.sa2

## Additional files

### Supplementary files
• Transparent reporting form

### Data availability
All data generated or analysed during this study are included in the manuscript and supporting files.

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
