## [Decision Letter]

**Acceptance summary:**

In this manuscript the authors have identified a *H. polygyrus* excretory secretory product (HpBAR1) that binds to murine ST2 and blocks signaling. They also find a similar protein that binds human ST2. During the revision, the authors have done a significant amount of work to strengthen their findings and demonstrate in vivo relevance. Unfortunately, the in vivo experiments examining ST2 with *H. Polygyrus* had to be cut short due to current lab closings in response to the COVID-19 pandemic. Nonetheless, their preliminary findings suggest interesting avenues of subsequent study.

**Decision letter after peer review:**

Thank you for submitting your article "A helminth-derived suppressor of ST2 blocks allergic responses" for consideration by *eLife*. Your article has been reviewed by two peer reviewers, and the evaluation has been overseen by a Reviewing Editor and Tadatsugu Taniguchi as the Senior Editor. The following individual involved in review of your submission has agreed to reveal their identity: Elia Tait Wojno (Reviewer #1).

The reviewers have discussed the reviews with one another and the Reviewing Editor has drafted this decision to help you prepare a revised submission.

Summary:

In this manuscript by Vacca et al., the authors identify a protein found in *H. polygyrus* excretory secretory products (HpBAR1) that binds to murine ST2 and blocks signaling. They also find a similar protein that binds human ST2. While the reviewers were generally enthusiastic about the work, there were questions about the physiological relevance of this protein both during allergic and helminth-specific responses in vivo. In light of the fact that *eLife* expects revisions to be completed within two months, the reviewers consulted and narrowed down the minimal amount of work required for publication. These essential revisions are listed below.

Essential revisions:

1) The data do not establish that HpBARI suppresses Alternaria-induced allergic responses in vivo by binding ST2. To assess this, the authors could treat Alternaria-exposed mice with the C-terminal tagged HpBARI that does not bind ST2 or suppress IL-33-mediated responses efficiently, and then compare the effects of the sub-par protein with that of normal HpBARI. This or a similar experiment would reveal whether the ability of HpBARI to bind ST2 normally impacts it's suppressive effects during Alternaria exposure.

2) To make this finding more relevant, additional in vivo data would be helpful: They could perform a standard *H. polygyrus* infection followed by staining of ST2 on ILC2s and compare the levels to those in uninfected mice. It might be hard to detect in the small intestine, where ILC2s predominantly express the IL-25R, not ST2. However, it is possible that HpBARI also acts systemically and blocks ST2 on ILC2s at distant sites, i.e. in the lung, adipose, etc.

Reviewer #1:

The manuscript by Vacca, et al. identifies a protein found in *Heligmosomoides polygyrus* excretory secretory products (HpBARI) that binds to murine ST2, blocks IL-33 binding to ST2, and decreases severity of allergic airway inflammation in mice. They also identify a similar protein that binds to human ST2. The manuscript is well written and clear. These studies fall into the growing field interested in helminth product-based modulation of allergic type 2 inflammation and are of broad interest.

To fully support the authors' conclusions, there are 3 areas that require additional clarification or data collection.

1) The data do not establish that HpBARI suppresses Alternaria-induced allergic responses in vivo by binding ST2. The authors could treat Alternaria-exposed mice with the C-terminal tagged HpBARI that does not bind ST2 or suppress IL-33-mediated responses efficiently, and then compare the effects of the sub-par protein with that of normal HpBARI. This or a similar experiment would reveal whether the ability of HpBARI to bind ST2 normally impacts it's suppressive effects during Alternaria exposure.

2) The authors show a few readouts of Alternaria-induced type 2 inflammation that are impacted by HpBARI treatment, with a focus on effects on ILC2 responses (which is reasonable). However, an expanded analysis of ILC2s and other parameters in the Alternaria model would be helpful to fully understand effects of HpBARI, particularly because recent literature has shown that IL-33 can mediate specific context-dependent effects in ILC2s as regards ability to produce IL-5 vs. IL-13 (Schneider et al., 2019; Ricardo-Gonzalez et al., 2018). The authors can clarify by:

a) Showing IL-5 ICS from ILC2s in Figure 3 (this helps to compare with previous data published on the effects of HpARI as well from Osbourn et al., 2017)

b) Showing IL-33 levels in the lung homogenate in Figure 3 or Figure 5 and Figure 4

c) Showing histopathology of the lung in Figure 3 or Figure 5 and/or Figure 4.

3) The IFNγ assay in Figure 7F needs to be explained better. If the authors are proposing that HpBARI_Hom2 could be leveraged to develop new therapies for human allergy, it is unclear why they chose to test suppressive effects of HpBARI-Hom2 on human cells using an IL-33-elicited IFNγ production assay. The authors should justify this choice in the text. The authors also need to show data from control conditions in the assay without IL-33 to establish that the effects of HpBARI were mediated via IL-33 and not IL-12.

Reviewer #2:

The capacity of the model helminth *H. polygyrus* and its excretory-secretory (HES) products to modulate host immune responses is well established, but the mechanisms that underlie this suppression are less clear. Previous studies reported various ways, including interfering with the IL-33 signaling pathway. Here, Vacca et al. describe a new *H. polygyrus* protein, HpBARI, which is a constituent of HES. Their data convincingly demonstrate that HpBARI can bind to the mouse IL-33 receptor chain, ST2, thereby blocking the engagement of the receptor by the ligand IL-33 and the downstream activation of type 2 responses. They tested this using a series of biochemical, in vitro and in vivo assays, with all the relevant controls (ST2-/-, competition, etc). Overall, the study is well conducted and clearly written. It represents a solid biochemical characterization of this previously unknown *H. polygyrus*-derived protein. I do not have any major technical criticism.

However, the relevance of their finding for *H. polygyrus*-mediated suppression of type 2 immunity is completely unclear. Although they demonstrate the potential of this protein to act as suppressor of IL-33-dependent type 2 response, their experiments do not address at all whether this matters in a physiological setting when mice are infected with this helminth (see below for suggestions how to possibly do this). Notably, HES are produced in vitro by culturing adult worms in serum-free cell culture media. This environment is relatively artificial, and the extent to which composition of in vitro-produced HES resembles that of HES produced in their natural niche (lumen of the small intestine), is unclear. Is HpBARI expressed at all under natural conditions, and if so, does it cross the epithelial barrier to block ST2 locally or even systemically to inhibit type 2 responses? Given that HES contains many worm-derived immunosuppressants, it would at some point be necessary to address their relevance. Is there any redundancy or does one of these factors dominate? These studies are very challenging and clearly not the scope of this study. However, without knowing if HpBARI is actually produced in vivo and whether it binds ST2 under those conditions, the relevance of their finding is unclear to me. Although this paper on a technical level clearly fulfills the requirements for publication in *eLife*, I am unconvinced that it provides sufficient advance that would make it interesting for a wider scientific community beyond the field of parasitology, particularly given that it is not the first report to demonstrate *H. polygyrus*' interference with the IL-33 signalling pathway.

1) Rigorous assessment of the relevance of this inhibitory mechanism in a physiological setting would require worm transgenesis, which, to my knowledge, has not been done successfully. However, an alternative way may be to target and neutralize these worm-derived secreted proteins using antibodies or nanobodies. Perhaps the authors have access to such tools? These could also be helpful for the detection of the HpBARI protein in vivo and to assess the degree of binding to ILC2s under physiological conditions by performing an ex vivo staining for ILC2-bound HpBARI following conventional infection with *H. polygyrus*.

2) To make this finding more relevant, additional in vivo data would be helpful: They could perform a standard *H. polygyrus* infection followed by staining of ST2 on ILC2s and compare the levels to those in uninfected mice. It might be hard to detect in the small intestine, where ILC2s predominantly express the IL-25R, not ST2. However, it is possible that HpBARI also acts systemically and blocks ST2 on ILC2s at distant sites, i.e. in the lung, adipose, etc.

Currently, this sentence in the Abstract is entirely speculative: "Thus, via HpBARI and HpARI, *H. polygyrus* inhibits both the IL-33 cytokine and its receptor, highlighting the importance of the IL-33 pathway in parasitic infection, the centrality of the CCP domain family to *H. polygyrus* immunomodulation,…".

3) The mechanism of blockade is not studied. Is the ST2 protein still on the surface, or is the loss of detection caused by receptor internalization? The authors could stain for intracellular ST2. Also, did the authors test other anti-ST2 antibody clones, which could differ between epitope blocking and loss of surface protein? Does the interaction between ST2-Fc and HpBARI block the binding of the anti-ST2 antibody to ST2 in their ELISA assay?

---

## [Author Response]

Essential revisions:1) The data do not establish that HpBARI suppresses Alternaria-induced allergic responses in vivo by binding ST2. To assess this, the authors could treat Alternaria-exposed mice with the C-terminal tagged HpBARI that does not bind ST2 or suppress IL-33-mediated responses efficiently, and then compare the effects of the sub-par protein with that of normal HpBARI. This or a similar experiment would reveal whether the ability of HpBARI to bind ST2 normally impacts it's suppressive effects during Alternaria exposure.

We have carried out additional experiments using Alternaria allergen, co-administrating highly active N-terminal tagged HpBARI, C-terminal tagged HpBARI (abrogated activity) or a control protein (HpAChE) also secreted by the parasite and produced in the same expression system (New Figure 3—figure supplement 1).

To clarify, the C-terminal tagged version of HpBARI does still have ST2-binding activity, as shown in the original submission Figure 6—figure supplement 1G (resubmission Figure 7—figure supplement 2), albeit with a much increased off-rate compared to N-terminal tagged HpBARI. In in vitro assays this limited binding activity could not replicate the suppression of IL-33 responses seen with N-terminal-tagged HpBARI at the concentrations used. However, in vivo some activity was still present, albeit much reduced compared to N-terminal tagged HpBARI. We found that while C-terminal tagged HpBARI tended to reduce eosinophilia, ILC2 responses and ST2 detection in vivo, none of these effects were as profound as those achieved by the N-terminal tagged version of the construct. Control protein administration did not change any of the results measured.

These results have been added to the manuscript as new Figure 3—figure supplement 1 and 2 and in the text subsection “HpBARI suppresses Alternaria-induced type 2 immune responses in vivo”.

2) To make this finding more relevant, additional in vivo data would be helpful: They could perform a standard H. polygyrus infection followed by staining of ST2 on ILC2s and compare the levels to those in uninfected mice. It might be hard to detect in the small intestine, where ILC2s predominantly express the IL-25R, not ST2. However, it is possible that HpBARI also acts systemically and blocks ST2 on ILC2s at distant sites, i.e. in the lung, adipose, etc.

We thank the reviewer for this suggestion, and have carried out an additional experiment, assessing levels of ST2 detection on cells from day 7 *H. polygyrus*-infected mice (new Figure 6, and Figure 6—figure supplement 1). These data are very compelling, and will lead to further studies on the kinetics, systemic effects and outcomes of ST2 suppression during *H. polygyrus* infection.

Unfortunately, these data are presented as the results from a single experiment. We had begun a repeat of this infection, but the government-imposed lockdown in response to the Covid19 outbreak meant that these mice had to be culled prematurely, with no data collected. We have accurately stated the number of replicate mice from the single experiment performed in the legend to Figure 6.

Our data show that in *H. polygyrus*-infected mice, ST2 detection was potently reduced on peritoneal lavage and lung ILC2s. We also observed a stark ablation of ST2 signal on a highly granular lineage-negative peritoneal population (most likely mast cells). As the reviewer correctly points out, ST2 is very poorly expressed in the intestinal environment, and no change in ST2 signal was detected in ILCs from the mesenteric lymph nodes (Figure 6—figure supplement 1A-C).

Of note, *H. polygyrus* miRNA-containing extracellular vesicles have also been suggested to suppress ST2 transcription, when delivered directly to the lungs (Buck et al., 2014). Our revised Discussion addresses this point and makes it clear that both extracellular vesicles and HpBARI may contribute to the reduced ST2 expression that we report.

Future studies will assess the extent, persistence and systemic effects of ST2 suppression in combination with anti-HpBARI staining (see response to reviewer 2, comment 1 below), however these experiments are outside the remit of this study, and now have to wait for the reopening of our laboratories.

Reviewer #1:The manuscript by Vacca, et al. identifies a protein found in Heligmosomoides polygyrus excretory secretory products (HpBARI) that binds to murine ST2, blocks IL-33 binding to ST2, and decreases severity of allergic airway inflammation in mice. They also identify a similar protein that binds to human ST2. The manuscript is well written and clear. These studies fall into the growing field interested in helminth product-based modulation of allergic type 2 inflammation and are of broad interest.To fully support the authors' conclusions, there are 3 areas that require additional clarification or data collection.1) The data do not establish that HpBARI suppresses Alternaria-induced allergic responses in vivo by binding ST2. The authors could treat Alternaria-exposed mice with the C-terminal tagged HpBARI that does not bind ST2 or suppress IL-33-mediated responses efficiently, and then compare the effects of the sub-par protein with that of normal HpBARI. This or a similar experiment would reveal whether the ability of HpBARI to bind ST2 normally impacts it's suppressive effects during Alternaria exposure.

See above

2) The authors show a few readouts of Alternaria-induced type 2 inflammation that are impacted by HpBARI treatment, with a focus on effects on ILC2 responses (which is reasonable). However, an expanded analysis of ILC2s and other parameters in the Alternaria model would be helpful to fully understand effects of HpBARI, particularly because recent literature has shown that IL-33 can mediate specific context-dependent effects in ILC2s as regards ability to produce IL-5 vs. IL-13 (Schneider et al., 2019; Ricardo-Gonzalez et al., 2018). The authors can clarify by:a) Showing IL-5 ICS from ILC2s in Figure 3 (this helps to compare with previous data published on the effects of HpARI as well from Osbourn et al., 2017)b) Showing IL-33 levels in the lung homogenate in Figure 3 or Figure 5 and Figure 4c) Showing histopathology of the lung in Figure 3 or Figure 5 and/or Figure 4.

We thank the reviewer for these comments, and have added citations of the Schneider and Ricardo-Gonzalez studies. In response to the specific suggestions:

a) As in our previous studies (Osbourn et al., 2017; and McSorley et al, 2014) we found that IL-33 blockade suppressed both IL-5 and IL-13 from ILC2s, but that this effect was more profound on IL-13 production, as even ILC2s from naïve mice produced a strong background IL-5 signal after PMA/Ionomycin/brefeldinA restimulation ex vivo. We have now added further data to address this point, with IL-5 production by ILC2s (and suppression by HpBARI) shown in new Figure 3—figure supplement 1. The context-dependence of IL-33 responses in vivo is clear, with low ST2 expression in the intestinal milieu (see new Figure 6—figure supplement 1A-C), however our data from *H. polygyrus* infection (new Figure 6) implies that HpBARI’s effects may be systemic.

b) Unfortunately, we could not carry out IL-33 ELISAs on lung homogenates prior to covid-19-related shutdown of our laboratory. We had planned to carry out a batch of these ELISAs on all in vivo samples (including those from new data shown in new Figure 3—figure supplement 1), but could not complete these measurements.

c) Finally, we have not taken samples for histopathology of the lung from the Alternaria experiments shown. At 24h after Alternaria administration, our previous data has not shown major histological changes in the lung, therefore it is difficult to score for suppression of these responses. In the 17-day Alternaria-OVA model we see major inflammation by histology, which is reduced on suppression of the IL-33 pathway (McSorley et al., 2014 and Osbourn et al., 2017), however this scoring was not repeated in the study presented here.

3) The IFNγ assay in Figure 7F needs to be explained better. If the authors are proposing that HpBARI_Hom2 could be leveraged to develop new therapies for human allergy, it is unclear why they chose to test suppressive effects of HpBARI-Hom2 on human cells using an IL-33-elicited IFNγ production assay. The authors should justify this choice in the text. The authors also need to show data from control conditions in the assay without IL-33 to establish that the effects of HpBARI were mediated via IL-33 and not IL-12.

We apologise if the IFN-γ assay results were unclear. We have now clarified this in the text. This assay is a very robust bioassay for human cell responses to IL-33. It depends upon IL-12 stimulation upregulating expression of ST2 on PBMCs, which, when stimulated with IL-33, then produce large amounts of IFN-γ. As the focus of this manuscript is to show effective blockade of IL-33 signalling, we felt that this bioassay was best suited to give a good titration of HpBARI activity on human cells. We have now added control conditions as requested, in new Figure 8—figure supplement 3, showing that both IL-33 and IL-12 are required for significant IFN-γ release, as has been previously published (Smithgall et al., 2008).

Reviewer #2:The capacity of the model helminth H. polygyrus and its excretory-secretory (HES) products to modulate host immune responses is well established, but the mechanisms that underlie this suppression are less clear. Previous studies reported various ways, including interfering with the IL-33 signaling pathway. Here, Vacca et al. describe a new H. polygyrus protein, HpBARI, which is a constituent of HES. Their data convincingly demonstrate that HpBARI can bind to the mouse IL-33 receptor chain, ST2, thereby blocking the engagement of the receptor by the ligand IL-33 and the downstream activation of type 2 responses. They tested this using a series of biochemical, in vitro and in vivo assays, with all the relevant controls (ST2-/-, competition, etc). Overall, the study is well conducted and clearly written. It represents a solid biochemical characterization of this previously unknown H. polygyrus-derived protein. I do not have any major technical criticism.However, the relevance of their finding for H. polygyrus-mediated suppression of type 2 immunity is completely unclear. Although they demonstrate the potential of this protein to act as suppressor of IL-33-dependent type 2 response, their experiments do not address at all whether this matters in a physiological setting when mice are infected with this helminth (see below for suggestions how to possibly do this). Notably, HES are produced in vitro by culturing adult worms in serum-free cell culture media. This environment is relatively artificial, and the extent to which composition of in vitro-produced HES resembles that of HES produced in their natural niche (lumen of the small intestine), is unclear. Is HpBARI expressed at all under natural conditions, and if so, does it cross the epithelial barrier to block ST2 locally or even systemically to inhibit type 2 responses? Given that HES contains many worm-derived immunosuppressants, it would at some point be necessary to address their relevance. Is there any redundancy or does one of these factors dominate? These studies are very challenging and clearly not the scope of this study. However, without knowing if HpBARI is actually produced in vivo and whether it binds ST2 under those conditions, the relevance of their finding is unclear to me. Although this paper on a technical level clearly fulfills the requirements for publication in eLife, I am unconvinced that it provides sufficient advance that would make it interesting for a wider scientific community beyond the field of parasitology, particularly given that it is not the first report to demonstrate H. polygyrus' interference with the IL-33 signalling pathway.

We thank the reviewer for their incisive and well-informed comments on this study. We agree that a focus on the role of HpBARI on the parasitic infection would be extremely interesting, and something which will be a subject of future studies. This study, however, was designed to identify novel immunomodulators from parasite secretions, and assess their potential for human use. We further address the reviewer’s specific comments on the role of HpBARI in *H. polygyrus* infection, below.

1) Rigorous assessment of the relevance of this inhibitory mechanism in a physiological setting would require worm transgenesis, which, to my knowledge, has not been done successfully. However, an alternative way may be to target and neutralize these worm-derived secreted proteins using antibodies or nanobodies. Perhaps the authors have access to such tools? These could also be helpful for the detection of the HpBARI protein in vivo and to assess the degree of binding to ILC2s under physiological conditions by performing an ex vivo staining for ILC2-bound HpBARI following conventional infection with H. polygyrus.

We absolutely agree with the reviewer, both that these would be fascinating studies, and, sadly, that the lack of transgenesis in parasitic nematodes is a continuing stumbling block to this type of research. The reviewer is correct that no transgenesis has yet been achieved in *H. polygyrus*. We agree that another route for assessment of the role of HpBARI would be the use of antibodies against the protein. We have produced polyclonal anti-HpBARI rat IgG, which effectively binds to recombinant HpBARI and a protein within HES (see new data in Figure 8—figure supplement 1). However, despite repeated attempts at optimisation, we could not achieve robust staining by using this reagent as a flow cytometry antibody. We are therefore planning to develop monoclonal antibodies against HpBARI, which could be used for in vivo blocking studies and flow cytometry staining, as the reviewer suggests. These experiments are part of our future plans, but the generation and selection of high affinity monoclonal antibodies can take up to 6 months, and so could not be completed within the timeframe of this paper.

2) To make this finding more relevant, additional in vivo data would be helpful: They could perform a standard H. polygyrus infection followed by staining of ST2 on ILC2s and compare the levels to those in uninfected mice. It might be hard to detect in the small intestine, where ILC2s predominantly express the IL-25R, not ST2. However, it is possible that HpBARI also acts systemically and blocks ST2 on ILC2s at distant sites, i.e. in the lung, adipose, etc.

See response above to essential revisions.

Furthermore, we agree with the reviewer that these studies are important, but that the suppression of ST2 in vivo could only be definitive if combined with staining for HpBARI on the surface of cells (see reviewer’s previous point above). This dual stain would allow the identification of those cells which had been affected by HpBARI, and those which express low levels of ST2. In the absence of useable antibodies for HpBARI, we have presented staining for ST2 alone during *H. polygyrus* infection (New Figure 6), and have discussed the strengths and limitations of these data in our Discussion section.

Currently, this sentence in the Abstract is entirely speculative: "Thus, via HpBARI and HpARI, H. polygyrus inhibits both the IL-33 cytokine and its receptor, highlighting the importance of the IL-33 pathway in parasitic infection, the centrality of the CCP domain family to H. polygyrus immunomodulation,…".

We have now reworded this part of the Abstract and a similar section in the Introduction to remove the speculative statements.

3) The mechanism of blockade is not studied. Is the ST2 protein still on the surface, or is the loss of detection caused by receptor internalization? The authors could stain for intracellular ST2. Also, did the authors test other anti-ST2 antibody clones, which could differ between epitope blocking and loss of surface protein? Does the interaction between ST2-Fc and HpBARI block the binding of the anti-ST2 antibody to ST2 in their ELISA assay?

These are very relevant questions, which we have attempted to address. We tried 2 different anti-ST2-APC clones, the first from ThermoFisher (clone RM-ST2-2 – used throughout the manuscript) and the second from Biolegend (clone DIH9) and found that binding of both was ablated. This is now shown in Figure 7—figure supplement 1B.

Unfortunately, our attempts at staining for intracellular ST2 were unsuccessful: fixation and permeabilisation appeared to ablate anti-ST2 antibody binding. We believe the simplest and most likely mechanism is that which is presented here, that HpBARI binds and blocks ST2, preventing ligation of the receptor by the cytokine.